# The Pioneering Role of *Sci* in Post Publication Public Peer Review (P4R)

**Ahmad Yaman Abdin** [1,2] , **Muhammad Jawad Nasim** [1] , **Yannick Ney** [1] **and Claus Jacob** [1,*]

1 Division of Bioorganic Chemistry, School of Pharmacy, Saarland University, D-66123 Saarbruecken, Germany; yaman.abdin@uni-saarland.de (A.Y.A.); jawad.nasim@uni-saarland.de (M.J.N.); yannick.ney@uni-saarland.de (Y.N.)
2 Univ. Lille, CNRS, Centrale Lille, Univ. Artois, UMR 8181–UCCS–Unité de Catalyse et Chimie du Solide, F-59000 Lille, France
* Correspondence: c.jacob@mx.uni-saarland.de; Tel.: +49-681-302-3129

**Abstract:** Scientists observe, discover, justify and eventually share their findings with the scientific community. Dissemination is an integral aspect of scientific discovery, since discoveries which go unnoticed have no or little impact on science. Today, peer review is part of this process of scientific dissemination as it contributes proactively to the quality of a scientific article. As the numbers of scientific journals and scientific articles published therein are increasing steadily, processes such as the single-blind or double-blind peer review are facing a near collapse situation. In fact, these traditional forms of reviewing have reached their limits and, because of this, are also increasingly considered as unfair, sloppy, superficial and even biased. In this manuscript, we propose forms of post-publication public peer review (P4R) as valuable alternatives to the traditional blind peer review system. We describe how the journal *Sci* has explored such an approach and provide first empirical evidence of the benefits and also challenges, such a P4R approach faces.

**Keywords:** commenting options; open access publishing; open peer review; public assessment; reward system; *Sci*; volunteers



## 1. Introduction

Modern theories of science describe the scientific process of discovery as a human endeavor, similar to other human activities, such as cooking or painting. This anthropocentric view of science has several implications, which in many ways also contradict our more traditional view of science as an attempt to somehow find the rules and laws hidden in nature and beyond. Rather than inching closer and closer to such hidden treasures already present in nature, scientists are actually busy constructing nature for us, in a pursuit amenable to human senses and concepts. Interestingly, such a constructivist approach brings together the processes of discovery, justification, and notably, dissemination. In fact, discoveries which go unnoticed have no or little impact on the scientific process. This rather surprising and bold statement is reminiscent of George Berkeley (1685–1753) and his famous tree and has also coined the rather nasty phrase of "publish or perish" as the bedrock of modern research [1,2]. Further inspection into the current dissemination practices brings us to the gold standard of scientific dissemination referred to as the peer review process. According to this constructivist strategy, and reflecting the appraisal of a dish or a painting in our comparison, a thorough review of a given manuscript by a handful of peers before publication directly and proactively contributes to the quality of the scientific article and hence the discovery presented therein. It is also assumed that this *a priori* scrutiny enhances the advancement of the relevant scientific field. In practice, scientists subsequently refine their research according to the review reports and revise their manuscripts accordingly, as failure to comply may result in the rejection of the manuscript by the journal. In practice, therefore drawing a firm line between the context of discovery,

the context of justification and the context of dissemination is unattainable, especially in the constructivist model and in light of the present system of academic publishing and peer review [3]. In contrast, fake discoveries published in respected journals or platforms may seriously spoil scientific practice and its integrity [4–6]. Hence, peer review serves as an effective tool to safeguard against low quality and fraud, yet it also needs to be open and fair, as any overly restrictive or biased approach may suppress and therefore equally damage good science and deprive it of its food and art necessary to blossom.

In this manuscript, we shall describe our experiences as editors, authors and reviewers with post-publication public peer review (P4R) as implemented in the MDPI open access journal *Sci* (ISSN 2413-4155) [7–10]. The pursuit of a more open and transparent alternative to the traditional peer review was motivated, firstly, by reports in the literature underscoring the flaws of single- and double-blind approaches and, secondly, as a consequence of our own publishing experiences during the last decades [11–16]. In Section 2, we therefore provide a brief historical review of modern scientific dissemination practices with a focus on peer review, thus highlighting the need for efficient safeguarding policies on transparency and fairness in scientific publishing. In Section 3, we contrast several such established and also suggested models of scientific publishing regarding their openness and transparency. We argue that the impact of the traditional single- and double-blind peer review processes on original work is often questionable and unfair. In Section 4, we discuss alternative publishing models, namely the self-publishing (SP) model. As part of this discussion, we recount our own pursuit in PurplePublishing.org. In the subsequent section, we present the P4R workflow and highlight its advantages and some of its shortcomings. We then present the P4R Hybrid model, which is also at the center of the dissemination strategy adapted by *Sci* since November 2020. We conclude in Section 6.

## 2. A Brief Look at the History of Modern Scientific Publishing

Indeed, peer review is not a novel innovation, as the origins of this strategy can be found in the early days of modern scientific publishing in the middle of the 17th century [17]. In fact, the landscape of scientific communication dates to antiquity, and since then has undertaken many significant changes, often fueled by the introduction of modern technologies [18]. One of the most decisive inventions in the history of modern scientific communication has been the introduction of the printing press by Johannes Gutenberg (1400–1468) in the 15th century, replacing the tiresome copying of manuscripts by hand with a pimped grape juice press. Arguably, the early 16th century had few "scientific discoveries" to communicate, as most of the sciences we cherish today had not been established by then and indeed the scientific form of communication with journals only entered the stage with the foundation of the first scientific societies in the 17th century, such as the Académie Française in 1635, the Deutsche Akademie der Naturforscher Leopoldina in 1652, the Royal Society of London in 1660, and Academie des Sciences in Paris in 1666 [19]. In the lee of these new societies, the first scientific journals were also established, for instance the *Journal de Sçavans*, translatable to "the journal of the wise or knowledgeable", whose first issue was published in January 1665 and the first journal of the Royal Society of London, the now famous *Philosophical Transactions*, which published its first issue just two month later [20,21].

Interestingly, these societies were keen to, and in a position to, safeguard the quality of their journals by a crude version of peer review, first introduced on a voluntary basis by the first editor-in-chief of the *Philosophical Transactions*, Henry Oldenburg (1619–1677) in 1665 [17]. In the 18th century, the number and periodicity of scientific journals increased, with 422 journal titles appearing between 1750 and 1790 [19]. The 19th century witnessed numerous more specialized disciplinary journals entering the fray, and the most prestigious journals of today, such as *Nature* and *Science*, date to this period. *Nature* was first published in 1869 by Norman Locker in London, and *Science* commenced in 1880 under the auspices of John Michels and Thomas A. Edison in New York [22,23]. The 20th century incorporated new technologies, which drastically influenced scientific communication in the form of

print and digital texts, leading eventually to the first examples of online open access journals such as *Psycoloquy* in 1990 and the *Journal of Medical Internet Research* (JMIR) in 1999 [24,25]. Subsequently, three main events inaugurated the advent of open access publishing in the 21st century, namely the Budapest Open Access Initiative in 2002, the Bethesda Statement on Open Access Publishing in 2003, and the Berlin Declaration on Open Access to Knowledge in the Sciences and Humanities, also in 2003 [26–28]. Indeed, since the turn of the millennium, journals in the fields of science, technology, and mathematics (STM) have turned into a major industry comprising more than 100,000 employees worldwide and an estimated annual revenue of 25.7 billion USD in 2017 [29]. As for scientific output, about 33,100 English-language peer-reviewed journals and 9400 non-English journals collectively published more than 8000 articles per day in 2018, amounting to a staggering 3 million peer-reviewed publications in STM in that year [29]. This number is likely to explode, as 15,335 open access journals are currently registered in the Directory of Open Access Journals [30]. In parallel, the burden caused by the peer review system is also increasing drastically, as researchers have reported investing around 70 million hours in sum for peer reviewing per year already [31]. And whilst more and more publishers are shifting from the traditional style of publishing bound journals paid for by the subscription fees of the readers to open access online publishing paid for by the article processing charges (APCs) of the authors, the traditional peer review process has more or less stayed the same, despite such technical improvements.

## 3. The Traditional Peer Review System

In fact, the approach to review manuscripts *a priori*, before they are published, has not changed that much since the days of Henry Oldenburg, although it is now more international and considerably faster. Despite its long tradition, this system of single- and double-bind review has several flaws, and today, confronted with millions of submissions each year, is pushed to its limits and likely to collapse soon [11–15]. Table 1 highlights some of the most prominent differences between the open access scientific publishing models available with regard to their openness [32]. Indeed, as one may notice from this list, there are numerous problems with the peer review system of today.

**Table 1.** Different models of open access scientific publishing in the context of public openness [32]. Please note that in the context of publishing online, the term open is a homonym which may refer to open access for readers, public access to reviews, open disclosure of the identity of the reviewers, and in P4R, also the ability of the public to comment openly on a given manuscript. Combinations of openness are possible and desired.

| | Traditional Single Blind | Traditional Double Blind | Hybrid Single Blind | Hybrid Double Blind | P4R | P4R Hybrid |
|---|---|---|---|---|---|---|
| Post-publication peer review | No | No | Optional [1] | Optional [1] | Yes [2] | Yes [3] |
| Interaction with unreviewed publication | No | No | Depends [4] | Depends [4] | Yes [5] | Yes [5] |
| Open identities of authors | Yes | No | Yes | No | Yes | Yes |
| Open identities of reviewers | No | No | No | No | Yes | Optional [6] |
| Open review reports | No | No | Optional [7] | Optional [7] | Yes | Optional [7] |
| Open accepted version of publication | Yes | Yes | Yes | Yes | Yes | Yes |
| Open interaction with accepted publication | No | No | No | No | Yes | Yes |

[1] Authors are offered the possibility to share their submission on a preprint platform. [2] Submission is immediately available on the journal platform prior to peer review. [3] Submission is immediately available on a preprint platform. [4] Depends on whether the authors choose to submit to preprint platform or not. [5] Interested parties from the scientific community can interact with the submission in the form of commenting, endorsing, and re-using. [6] Reviewers can choose whether or not to sign their review reports. [7] Authors may choose to render the review reports and their responses public and accessible alongside their publication.

Firstly, and perhaps most obviously, the high demand for fast and quality reviews by expert scientists who are not paid for their services is impossible to meet, and sloppy reviews, often by less interested and/or experienced colleagues, is just one consequence. A recent survey has exposed this lack of motivation among senior scientists to engage in this style of peer review and has highlighted a tendency to deflect the task of peer review in the direction of their subordinates [33]. Notably, some respondents to this survey have also commented on the issue of ghost-writing peer reviews as unethical, undoubtedly another

common shocking practice and major flaw in this process. In contrast, colleagues who may be genuinely interested to review a given scientific piece, and who may indeed be suited to comment and to provide a valuable contribution, are excluded from this process, which *de facto* takes place behind closed doors. This applies especially to younger and less prominent colleagues and also to some extent to colleagues from less affluent countries. Traditional reviewing is therefore in some respects also plainly discriminatory, and it is necessary to spell this out once and clearly.

Secondly, as the evaluation of a manuscript and its impact takes place behind closed doors, often by a maximum of three peers, such secret reviews should be considered unacceptable in a modern global society aiming at openness and transparency. A few quite interesting examples highlighting the anthropocentric subjectivity involved in such processes can be found in the literature and are likely to be just the tip of the iceberg [34,35]. This issue of authoritarian rather than just authoritative subjectivity also needs to be stated clearly, especially if a constructivist approach is taken which considers such scientific activities embedded in the respective civil societies of the day.

This traditional system of reviewing manuscripts is therefore not only sloppy, yet also highly secretive and subjective. In other words, the double-blind peer review, by some heralded as the best invention since sliced bread, is not a great fortune for open and fair scientific dissemination; it is actually in some respects a societal anachronism and disaster for research. And whereas an open peer review approach may avoid some of this criticism, it is also conducted *a priori* and exclusively by a handful of selected reviewers, and hence may result in acceptance of mediocre, and rejection of excellent, manuscripts. In other words, by considering the apparent significance and impact of the content of a manuscript, such peer review and reviewers clearly overstep another red line, as the true impact of a given piece of research on science is simply unpredictable, and can only be judged years after the publication, in other words, *a posteriori* [36,37].

Fourthly, as peer reviewers often demand major changes to a given manuscript, which the authors are required to carry out, one needs to ask if this does not infringe on the originality of the authors and their work [38]. Indeed, requests made by reviewers to change manuscripts as a precondition of publication are often considered by authors as unwanted and also unwarranted interferences with their piece of work. Letters of rebuttal rather than polite replies have become quite explicit on this in recent years, and the matter here is not that simple. This question is seldomly asked, yet important, as a manuscript is not only a piece of science; it is also a piece of personal writing and hence art authored by the persons who probably have the most experience with the matter at hand. Neither William Shakespeare nor Joanne Rowling have been told to resubmit their writings to this or a different journal or to carry out major revisions within ten days of receiving the reviews. Although the comparison is daring, just take a bunch of referees asking Beethoven to correct his symphony before it can be performed publicly, and you may hear a few notes of dissent you may not forget. Most journals simply brush this matter of personal freedom of expression free of external and anonymous interferences aside, although it is certainly a bread and butter issue for many authors and therefore worth addressing.

Indeed, this very manuscript aiming at the protection of the originality of authors and their style also has its unique style and, rather astoundingly, has therefore attracted a few comments during the review process suggesting changes to this style. For more information, please refer to the review reports of this article.

Not surprisingly, there are these and numerous other smaller and also larger complaints about the current reviewing process by frustrated authors and equally frustrated reviewers, who together often consider the traditional approach as ineffective, discriminatory, tedious, difficult, biased, bothering, and undemocratic [39].

## 4. Alternative Ways of Publishing

It is therefore not surprising that authors have looked for alternative ways to publish their scientific data [40–45]. One possibility, of course, is the method of self-publishing (SP)

by simply placing manuscripts on the website of the institute or university. Although this sounds rather unorthodox, self-publishing has been the method of choice for centuries, and terms of some of the traditional publishing houses, such as Oxford University Press (OUP), hint at the origin of this method. So indeed, why not collect the manuscripts produced by your university and place them on your university website? Technically, this strategy of SP is fairly simple and easy today, and certain platforms, such as ArXiv, BiorRxiv, ChemRxiv, and F1000 practice this already [46,47]. Nonetheless, it also has its pitfalls [48,49]. Besides the need to safeguard against theft by rather cumbersome registration and archiving methods, such as Crossref and Clarivate, self-publishing also lacks the seal of approval by independent referees and, in any case, esteem and publicity [50,51]. We have explored this model ourselves for a few months and, faced with these issues, have abandoned it very swiftly. In essence, you cannot tweet your science like a recent US president has been tweeting his policies, firstly because no one may accept it as being correct, and secondly because you are not in the Oval Office.

In order to maintain a minimum level of scientific quality and validation, a more sophisticated strategy should therefore combine SP with certain elements of independent endorsement. Once again, this is possible technically, as commenting options are easily included in websites and used widely on sales platforms such as Amazon and eBay. This avenue has led us to a project referred to as Purple Publishing [52]. Here, self-publishing on a specific website is combined with a public commenting option so documents can be published free of any interference of referees, yet their quality can be commented upon openly. The result has been PurplePublishing.org, an online platform which offers social-media-style services for post-publication open and public commenting, or, if one may prefer to call it so, post-publication peer review. Similar attempts to popularize a culture of post-publication public peer reviews have already been explored by PubMed Commons in 2013, with disappointing engagement from the scientific community leading to its discontinuation in 2018 [53,54]. In retrospect, PubMed Commons may be considered as a valuable tool enabling scientists to leave public comments on manuscripts indexed on the US National Institutes of Health (NIH), albeit only for publications which had been peer reviewed already before publication. In the same context and year, Pubpeer was established as an online platform to post public and anonymous comments on already published manuscripts [55]. Anonymous commenting on this platform has been controversial [56,57].

Such public endorsement after publication combines modern features of scientific communication with those of a web-based marketplace to exchange and rate products, in this case manuscripts worldwide, and is entirely public, transparent, open, and democratic. Nonetheless, the purple style of SP is also faced with problems of registering and archiving manuscripts, and the question of if and how revisions may be carried out. As for any other model of SP, it is also notable that such a publishing website lacks standing within the scientific community, since it is not registered on any of the major search engines such as SciFinder or Medline and has no impact factor. In the case of PurplePublishing.org, visits have been sparse, and we have also for now shelved this alternative.

## 5. Post-Publication Public Peer Review (P4R) in *Sci*

Taking stock of these challenges associated with self-publishing, a combination of a traditional publishing strategy and publishing house with the beneficial aspects of post-publication public peer review looks more promising. In this case, the manuscript in question is handled by the editorial staff of a professional publishing house and published by an established online journal after a brief check for content and consistency. This check is simply unavoidable, so no unsound or improper reporting, plagiarism, or offensive material is being posted and hosted.

In theory, the manuscript is then open for public review by the entire community, and this open exchange between authors and reviewers is documented openly, as it is informative for the authors and readers and also useful for possible revisions. The originality of the authors, Beethoven or otherwise, is maintained; the review process is open and conducted

by colleagues who are indeed genuinely interested in reviewing; the comments, replies, improvements, and rebuttals are public; and, as an added extra, the reviewers and their valuable efforts are appreciated and no longer go unnoticed.

This approach towards publishing has been implemented in the MDPI journal *Sci* (ISSN 2413-4155), as illustrated in Figure 1. MDPI launched *Sci*, an open access journal which covers most fields of scientific research in March 2018 to provide transparency on the traditionally secret process of peer review, and subsequently adapted most aspects of the post-publication public peer review (P4R) strategy discussed here [58,59]. The P4R system in place from March 2019 until November 2020 promised authors immediate visibility of their manuscripts on the journal's online platform after a brief and limited check of scientific soundness and proper reporting and against plagiarism and offensive material, as discussed already. This check by the editorial office included, for instance, the quality and clarity of figures, the experimental design, and possibly the logic in the discussion and conclusions. In contrast, handling by the editorial office avoided traditional reviewing and no decisions based on issues such as scientific significances and impact were taken.

This P4R system introduced by *Sci* in 2019 removed many of the traditional and often subjective opinions on originality and significance by a nominated few, replacing this traditional way of reviewing by a democratic and transparent environment in which authors, reviewers and indeed the entire scientific community could interact openly. From this date, any scientific piece which fulfilled the basic requirements of a scientific manuscript was published in *Sci*. Its value, rather than being assessed by a handful of self-styled experts *a priori,* was then proven *a posteriori*. Furthermore, rather than relying on referees, the ratings and comments provided by readers and practitioners became available to provide an open and public seal of approval of quality, opening the door to new investigations and discoveries. These aspects of open and public review were designed to form the basis for a healthy publishing system based on what may be called crowd reviewing.

Although *Sci* tried to provide an answer to many of the complaints of many authors and reviewers mentioned in Section 3 by introducing this system of P4R, it also encountered a range of new problems once P4R was put into practice. As one of the authors of this article is also editor in chief of *Sci*, we shall discuss the most unexpected and burning issues with P4R encountered in the case of *Sci* here firsthand. Some of these difficulties, such as lengthy and uncoordinated reviews, were the outcome of P4R itself and directly related to the practice of open voluntary review. We must therefore consider some of these obstacles associated with P4R, mention the lessons learnt, and also highlight the differences between an open marketplace for daily items such as toilet paper on the one side and one for scientific dissemination on the other. As in the previous section, we shall be frank and direct here, as some issues need to be said openly and unmistakably.

Firstly, P4R is necessarily slower than the traditional review process employed by journals, since it requires potential volunteers to take notice of the manuscript, register as voluntary reviewers and then actually review manuscripts and post their reviews. In short, these potential reviewers need to come forward in sufficient numbers, register quickly, gain approval and comment, a cumbersome procedure which can take days if not weeks and months. In some instances, such volunteering has been promising when compared to traditional peer review, with very detailed, constructive and polite reviews received rather promptly. The ability of readers to comment publicly on manuscripts is definitely a valuable aspect of modern scientific exchange and indeed, since 2020 MDPI has introduced such commenting options on published articles to most of its journals, albeit not as part of the reviewing process itself.

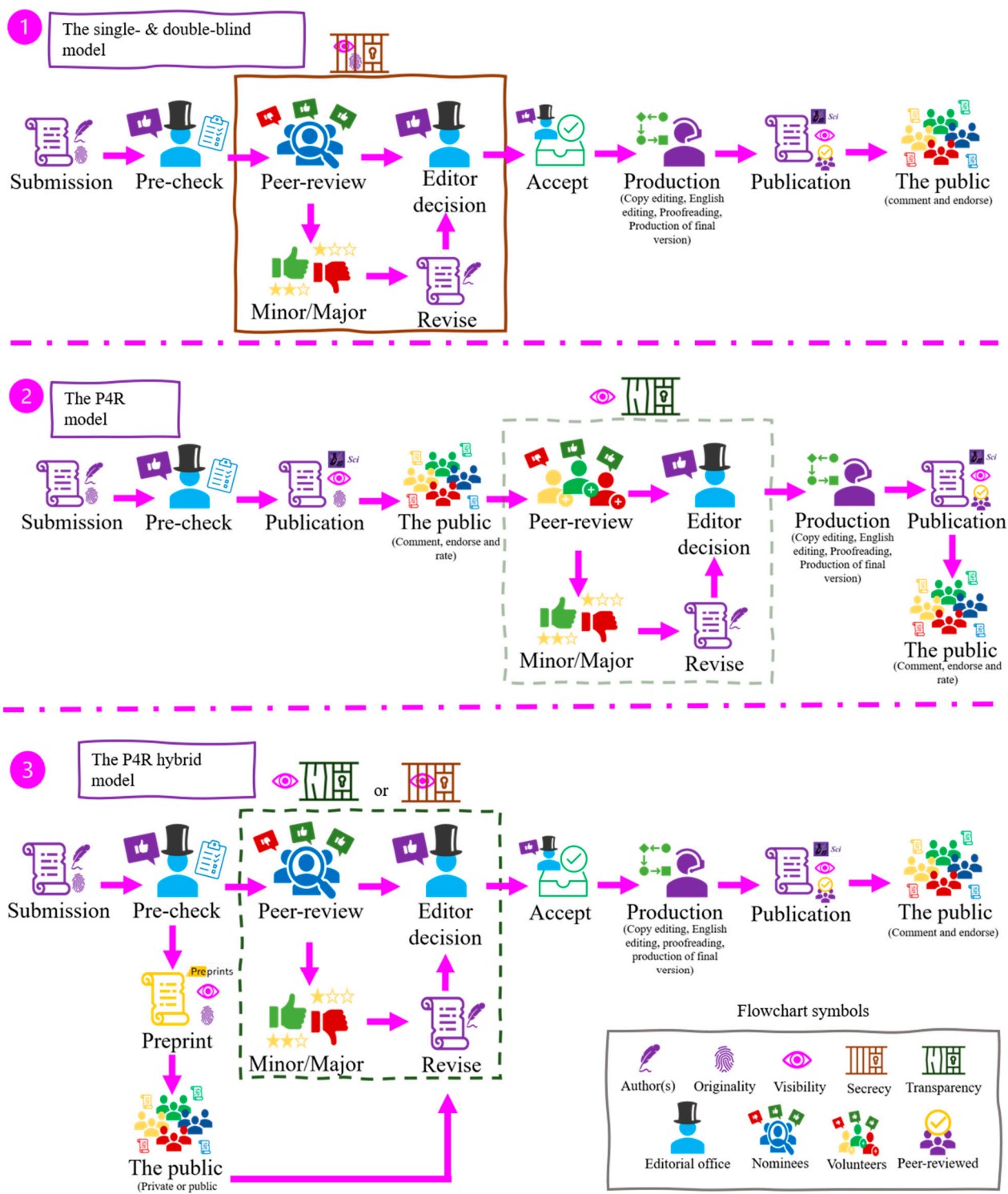

**Figure 1.** The traditional model of peer review relies on nominated reviewers who professionally review a given manuscript sent to them within a short period (Panel 1). The manuscript is then rejected or revised and published based on the reports of these nominated reviewers. In the P4R model, the manuscript is pre-checked to guard against inadequate form and content (Panel 2). It is also published rapidly as an original piece of science and art and is then open for public comments. Since revisions are difficult to enforce and result in multiple DOIs, a hybrid model involving two, rather than one, online platforms and journals has been proposed, in which volunteers and professional reviewers interact hand-in-hand after the manuscript is published on *Preprints* and before it is revised and published in *Sci* (Panel 3). The decision is taken based on a joint assessment, and once the manuscript is published on *Sci*, further commenting by the public and for longer periods, yet no refereeing or revision, is also possible.

In other cases, and we have to mention this, reviewing has been slow and suitable reviewers simply did not register, hence extending the process of reviewing and revision for months, and eventually leading to incomplete reviews or the need to choose and invite reviewers almost personally to register. In some instances, the resulting manuscript processing time (MPT) has been unacceptable. Such slow evaluations may be fine for the odd roll of toilet paper on Amazon; they are not for a scientific paper. Indeed, whereas readers may comment on existing articles in the literature for years, crowd reviewing does not have this luxury and must be fast and furious. If not, the process of revision may stall and the manuscript becomes stuck for months. We have witnessed such issues firsthand as authors of an article placed online for P4R on 12 March 2019 [60]. Rather than facing enthusiastic refereeing by readers, our manuscript was read and noted, yet not commented upon for more than 40 days, after which we received an email from the editorial office stating that while the manuscript is indeed online, no one from the scientific community has volunteered to review it and no feedback was received up to that date. It took until 26 June 2019, to receive the first round of three reviews. Although this may have been an unfortunate coincidence, another manuscript of ours submitted to *Sci* on 2 August 2019, was also stuck in review and only accepted on 1 February 2021, a total 16 months after submission [8]. Although reliable quantitative data on crowd reviewing is not yet available for *Sci*, these are not single incidences yet rather an issue with P4R itself.

Secondly, relying on volunteers rather than nominated experts is more open and democratic, yet also paves the way to low-quality reviews, bias and behind-the-scenes nominating activities by authors and the journal itself. Furthermore, not every volunteer is also qualified to comment, as may be the case for the toilet paper and this issue has been difficult to handle. It has occasionally resulted in inadequate reviews by inadequately qualified reviewers and almost personal clashes between these reviewers and the authors feeling treated inadequately. Again, as editor in chief we have dealt with several rather nasty examples we could cite here, usually involving authors complaining about the inexperience of referees and their refusal then to revise accordingly. As for the duration of P4R, the quality of refereeing is another issue one needs to resolve, although it is probably not limited to P4R and also present in traditional peer review as mentioned already.

Indeed, a certain refusal by authors to accept comments or reviews has been noted in *Sci*, possibly fueled by the fact that the manuscript had been published *de facto* already as part of the P4R strategy of post-publication review. The kind of rebuttal letters we have witnessed have been surprisingly rude and often personal.

In practice, the P4R model has also caused a small logistical mess, as the options of retraction or rejection are not really available in P4R, where a highly problematic public naming and shaming of a weak manuscript looks to be the only tool then available to guard against lack of quality. Whereas one may buy a roll of low-rated toilet paper and have fun with it, one needs to be considerably more careful if buying the content of a publication which has received major criticism by the community.

Fourthly, the authors who are keen to take the advice of their peers and happy to revise their manuscripts have swiftly noticed that the original versions and the revisions necessarily have to constitute separate publications with individual DOIs, requiring exuberant archiving of various versions and also an eloquent color-coding system to show readers the version accepted after revision.

Although P4R has not failed in *Sci* during 2019 and 2020, and these issues may in theory be resolvable technically, some of the limits of P4R have been reached in practice. This has also impacted on the standing of the journal within the scientific community. The journal *Preprints*, for instance, which was launched almost two years before *Sci* and shares the notion of post-publication peer review, received 11,304 submissions in 2018, 7248 in 2019, and 17,939 in 2020 until November 2020, of which 5690 were placed online in 2018, 4140 in 2019, and 9581 in 2020. In stark contrast, *Sci* received 83 submissions in 2018, 160 in 2019, and 62 in 2020, and has published just 7 articles in 2018, 56 in 2019, and 34 in 2020. These numbers are modest and clearly not as expected for a peer review model designed

to address the various and justified requirements and hopes of a large proportion of the scientific community. Whilst we accept that one possible issue faced by authors was the inability to include *Sci* as a P4R journal in Clarivate's Web of Science and Science Citation Index to date, whining and moaning about the traditional peer review system is one thing; manuring on a possible, albeit not perfect, alternative such as P4R and *Sci* to resolve these issues is quite another.

Based on the clear benefits and also problems associated with P4R, *Sci* has now adjusted its publishing model to combine the benefits of traditional reviews and P4R [61]. The resulting hybrid model is also added to Figure 1, Panel 3 and Table 1, Column 6. In practice, authors submitting manuscripts to *Sci* are offered the possibility of posting the submitted version at *Preprints* during the period the manuscript is sent for hybrid single-blind peer review, so P4R is possible then in theory [62]. *Preprints* itself is a multidisciplinary, open access, non-profit, cost-free online platform on which authors may indeed place their publication as open to and reusable by the community before peer review and publication in a peer-reviewed journal. This platform allows interested parties from within the scientific community to provide private and public comments on publications, similar to the marketplace mentioned before, and authors are offered the possibility of revising and submitting updated versions, each with individual DOIs. In parallel, a traditional, fast-track, and decisive review by nominated reviewers is performed, in which the reviews are public and the reviewers have the option to show their identity. Once completed, the version approved is then published in *Sci* with only one DOI. This strategy encompasses just two versions of the manuscript, one *a priori* to publication in the journal on the *Preprints* platform, and one after review in the journal *Sci* itself. Despite its own issues we cannot discuss here in detail, such as manuscripts getting stranded on *Preprints*, the *Preprint* plus journal strategy therefore avoids one of the main problems with P4R, namely that each version requires its own DOI. Indeed, distinguishing between different versions of the same manuscript on *Sci* has been cumbersome. As a further homage to the values of P4R, *Sci* allows an open discussion on its manuscripts *a posteriori*, after publication in *Sci*, and similar to P4R, yet without the option of further revisions, thereby avoiding long processing periods and, once again, the need to register each revised version with a specific DOI. In the case of this article, and rather disappointingly, not a single review or comment has been posted about the manuscript on *Preprints* in the first month of its publication, and accordingly, only the traditional review with nominated reviewers has taken place.

## 6. Conclusions

In summary, the introduction of online and open access publishing at the turn of the millennium has resulted in a rapid increase in the number of journals and articles published. Although this is laudable, since it fosters communication of science and between scientists, it has also pushed the traditional peer review system to its limits, raising several concerns linked to openness and fairness. Alternatives, such as self-publishing on personal websites or social media, are not really suitable for scientific communication, as these methods lack the seal of approval from the community and also the followers to manage a wider readership.

Public post-publication peer review (P4R) in journals provides a valuable alternative, yet also faces its own problems of registering volunteer reviewers, sluggish reviews, animosities between authors and reviewers, slow processes and logistic issues related to rejections, retractions, archiving and inflated DOIs. The journal *Sci* has experimented with some of these aspects of P4R and after learning some valuable lessons during 2019 and 2020, has adopted a new policy of P4R, which provides ample opportunities for the public to comment during and after the episodes of reviews and revisions on *Preprints* and in *Sci*.

*Sci* is therefore now in a good position to explore further some of the major concerns of the authors and reviewers, from maintaining originality and the need for an open and transparent review to the ability of the scientific community to have its say on the content and quality of a piece of science published in this special journal. *Sci* also strives for

such reviewers to get noticed as important contributors to science. In order to improve P4R further, the scientific community itself, rather than simply complaining about the traditional peer review processes, needs to take a more proactive role in volunteering to review manuscripts which are of profound interest professionally, just as in any good marketplace. This involvement of the wider community is clearly essential for the success of P4R and is a prerequisite for any journal to replace the traditional closed-door reviewing by nominated reviewers with a truly open access. Indeed, besides granting open access to read articles, reviews and the names of reviewers should also be in the open, as should be the door for anyone from the community to comment and become engaged in reviewing. In other words, if one complains about the traditional peer review processes, and is offered a valuable alternative which opens doors, then one should also take it.

The next couple of years should provide further evidence of the revolution we are witnessing in online and open access publishing. They will also show how *Sci* and P4R can be improved further to stay at the forefront of modern-day publishing, assuming that the community demanding such a modern way of public reviewing and publishing has and plays the balls necessary to succeed.

**Author Contributions:** C.J. wrote, validated, and supervised this study. A.Y.A. conceptualized, wrote, visualized, and carried out the necessary investigations of this study. M.J.N. and Y.N. wrote and helped visualize the study. All authors have read and agreed to the published version of the manuscript.

**Funding:** The authors appreciate the financial support provided by Landesforschungsfoerderungsprogramm of the State of Saarland (Grant No. WT/2—LFFP 16/01), and the INTERREG VA GR program (BIOVAL, Grant No. 4-09-21). The authors would like to express their explicit gratitude to all the members of the EU COST Action 16112 "NutRedOx" and for the support of the Erasmus+ programme of the European Union, Saarland University (SAARBRU01).

**Data Availability Statement:** No new data were created or analyzed in this study. Data sharing is not applicable to this article.

**Acknowledgments:** This article is one of the many efforts of the Pharmasophy Division, Saarland University. We would like to thank Michèle Friend, Sébastien Paul from UCCS, Centrale Lille, Lena Kaestner, and Klaus Ruthenberg, and all the colleagues from Pharmasophy and the Academiacs International network (www.academiacs.eu accessed on 11 March 2021) for their helpful discussions and advice. We would also like to thank Aimie Li, Delia Mihaila, Shu-Kun Lin, and Franck Vazquez from MDPI for their patience and continuous support. Finally, we would like to extend our deep gratitude to the reviewers for their input and orientation.

**Conflicts of Interest:** The authors declare no conflict of interest.

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
