# Peer review of "The Pioneering Role of Sci in Post Publication Public Peer Review (P4R)"

_publications, doi:10.3390/publications9010013_

Round 1
Reviewer 1 Report
The article describes an alternative type of peer review that has been tested in MDPI Sci journal and then adjusted after lessons learnt during its pilot phase.
The intrinsic interest of the article lies in the description of the so-called P4R and subtype Hybrid P4R workflows however the study is not sufficiently contextualized. For instance, its introduction and to a lesser extent its second and third sections are not fully linked to the sections that follow later, which are those that are the core of the study and actually describe the P4R model. Thus, it would be helpful to the reader to have a better context about P4R, not only as to how it falls within the landscape of current innovative peer review efforts in the scholarly communications system but also as to the motivations and goals by Sci Editorial Board when deciding on supporting such innovative peer review model. Likewise, the article does not refer in sufficient detail to other post publication peer review initiatives launched by either repositories and journals. Indeed, post publication peer review has been tried in different shapes several times before, with somewhat dissapointing results (for instance, PubMed Commons in 2013). On another front, some of the components identified by the authors as elements that have hampered rapid adoption of P4R, namely several challenges around reviewers and difficult DOI management of versions, are interesting enough to merit further elaboration in the paper.
The article needs revision as far as its style is concerned: for instance, sentences such as "In essence, you cannot tweet your science like Donald Trump has been tweeting his policies, firstly because no one may accept it as being correct, and secondly because you are not Donald Trump" or "In other words, the double-blind peer review, by some heralded as the best invention since sliced bread" do not really add anything relevant to the paper and are far from the standard scholarly language and evidence.
The article also includes some imprecise information, for instance, the first open access journals made their appearance well before the 21st century. By way of illustration Psycoloquy and the Journal of Medical Internet Research (JMIR) were launched in 1989 and 1998, respectively.
Author Response
Thank you for your comments. The manuscript has been revised. Please see attachment for authors' reply.

Reviewer 2 Report
This article describes a new form of peer review called publication public peer review (P4R) and a case study of P4R at the journal Sci. The article is well written and structured, I am impressed with the authors' insight on the limitations of the double-blind peer review system and their frankness on the weakness of P4R. It will be a valuable contribution to the literature of scholarly communication. I suggest that the article can be enhanced before accepted for publishing: 1. The editor decision-making process (what is the criteria and how to resolve author/reviewer conflicts). 2. More data from a case study of the journal Sci.
The major difference between P4R hybrid and double-blind peer review is that P4R will publish the manuscripts in the preprint server and collect volunteer comments/reviews from there (Figure 1). My experience with preprint is that it is hard to have readers submitting reviews/comments. I want to see the data showing how many comments collected from preprint and how many of the comments actually got solved/revised by the authors.
Pg 5, line 85. It should be "visits have" instead of "vists have"
Pg 7, line 266-269. I think it tries to describe the technical issue that every revision creating a new DOI. I suggest revising/re-organize the paragraph for a more easy read.
Author Response

(The authors gave the same response as above.)

Reviewer 3 Report
Dear Authors,
Your manuscript is the result of a description of the experiences you have gained during the period of publishing for a new scientific journal: Sci. The content of the work focuses mainly on issues related to various variants of peer review, and in particular on the reasons for the evolution of this process in the case of Sci. The subject matter is extremely interesting not only for researchers in article publishing, scientometrics and bibliography, but also for a much wider audience. The manuscript is clearly written, well composed and professionally prepared. The work has the right structure: from introductory chapters, through logical development of the issue to the conclusions.
General and discussion comments (I am asking for answers):
- The work could only be considered as editorial, but an important and innovative topic supported by own experience adequately justified the scientific nature of the manuscript. However, the article describes the phenomena and reality from the point of view of the Sci editorial staff (see line 185), so (in my opinion) there is a need to include information about the nature of the authors' relationship with the journal in the manuscript. On the Sci website I found information that Prof. Claus Jacob is the editor-in-chief of the journal. I propose to add in line 19 with: "We descibribe ..." the sentence that the authors are involved in the journal's publishing policy.
- Lines 272/273: It is unclear why there were such differences between the number of submissions and the number of articles published.
- Is the new method of peer review compliant with the requirements of bibliographic databases, eg those listed in lines: 276/277?
- I think it is worth mentioning in the manuscript that MDPI introduced the possibility of applying for a review at the beginning of December 2020. I have used this opportunity several times and believe that it improves the adaptation of the subject of the manuscripts to the scientific interests of the reviewers. This is confirmed by my colleagues, also involved in numerous reviews for various publishing houses.
- I have one more point to consider. I encourage the authors to support their valuable arguments with a quantitative approach. Maybe you can add comparisons with other MDPI publishing house journals in the same period?
- Additional comment: I remember the information that the Sci journal with a wide scope of aims & scope was launched in MDPI, but I did not "reach" the information about the innovative approach to peer review. Maybe one of the reasons for the difficulties described on lines 240-… is the lack of proper/wider promotion?
- Detailed comments:
The purpose of the work was not clearly indicated in the manuscript.
Line 57: There should be a reference at the end of the sentence.
Table 1 caption: please add reference (s)
Lines 168/169: Please consider removing the name and using a more generic description (president?). This article should be (in my opinion) apolitical, and the use of a particular politician's name may be perceived differently by his supporters and opponents.
Line 185: "in our case" means: "Sci journal"?
Reference 24: Journal name (Sci) is missing.
Author Response

(The authors gave the same response as above.)

Round 2
Reviewer 1 Report
The text has got improved with more contextual and background information that helps the reader focus on the discussion around the current flaws in the peer review system and on the pilot experience and lessons learnt by P4R model. Table 1 is placed in the wrong section.
I still strongly encourage authors to revise some phrases that sound too informal for a scholarly piece of work. Avoiding doing so may have undesirable consequences, namely, readers that are intrinsically interested in the topic may get somewhat dissapointed at some parallelisms proposed by the authors, which may reduce the credibility of the work.
Author Response
"The text has got improved with more contextual and background information that helps the reader focus on the discussion around the current flaws in the peer review system and on the pilot experience and lessons learnt by P4R model. Table 1 is placed in the wrong section."
We have corrected the location of Table 1 and added the manuscript to the new template of Publication.
"I still strongly encourage authors to revise some phrases that sound too informal for a scholarly piece of work. Avoiding doing so may have undesirable consequences, namely, readers that are intrinsically interested in the topic may get somewhat dissapointed at some parallelisms proposed by the authors, which may reduce the credibility of the work."
We would like to thank the reviewer for their effort and time. We highly appreciate their opinions and concerns. Nevertheless, and with all due respect, we would like to maintain the style and language adapted in this manuscript.
Reviewer 2 Report
The first table, at pg3, is apparently placed incorrectly. It should belong to the section 3 instead of section 2. I think it is ready to be published after correcting this issue.
Author Response
"The first table, at pg3, is apparently placed incorrectly. It should belong to the section 3 instead of section 2. I think it is ready to be published after correcting this issue."
We have corrected the location of Table 1 and added the manuscript to the new template of Publication.
We would like to take this chance to thank the reviewer once again for their effort and time
Reviewer 3 Report
Dear Authors,
Thank you very much for your answers to my doubts and for introducing changes to the content of the manuscript. I think this is a work that readers can learn a lot from. I am glad that I successfully applied for the opportunity to review it.
During the author's proofreading, pay attention to the correct location of Table 1.
And good luck in your efforts to develop the Open Access publishing system!
Author Response
We have corrected the location of Table 1 and added the manuscript to the new template of Publication.
We would like to take this chance to thank the reviewer once again for their effort and time